# Severe Posaconazole-Induced Glucocorticoid Deficiency with Concurrent Pseudohyperaldosteronism: An Unfortunate Two-for-One Special

**DOI:** 10.3390/jof7080620

**Published:** 2021-07-30

**Authors:** Alejandro Villar-Prados, Julia J. Chang, David A. Stevens, Gary K. Schoolnik, Samantha X. Y. Wang

**Affiliations:** 1Department of Medicine, Stanford University School of Medicine, Stanford, CA 94305, USA; avillarp@stanford.edu; 2Division of Endocrinology, Metabolism, and Gerontology, Stanford University School of Medicine, Stanford, CA 94305, USA; jchang89@stanford.edu; 3Division of Infectious Diseases and Geographic Medicine, Stanford University School of Medicine, Stanford, CA 94305, USA; stevens@stanford.edu (D.A.S.); gks007@stanford.edu (G.K.S.); 4California Institute of Medical Research, San Jose, CA 95128, USA; 5Division of Hospital Medicine, Stanford University School of Medicine, Stanford, CA 94305, USA

**Keywords:** coccidioidomycosis, posaconazole, hypokalemia, pseudohyperaldosteronism, 11β-hydroxylase

## Abstract

A 56-year-old Hispanic man with a history of disseminated coccidioidomycosis was diagnosed with persistent glucocorticoid insufficiency and pseudohyperaldosteronism secondary to posaconazole toxicity. This case was notable for unexpected laboratory findings of both pseudohyperaldosteronism and severe glucocorticoid deficiency due to posaconazole’s mechanism of action on the adrenal steroid synthesis pathway. Transitioning to fluconazole and starting hydrocortisone resolved the hypokalemia but not his glucocorticoid deficiency. This case highlights the importance of recognizing iatrogenic glucocorticoid deficiency with azole antifungal agents and potential long term sequalae.

## 1. Introduction

*Coccidioides* (two species) is a dimorphic fungus endemic to the United States and countries in Central and South America [1]. Individuals are usually exposed to *Coccidioides* by inhaling spores, and disease primarily manifests in the lungs [1]. In some instances, *Coccidioides* disseminates and infects the central nervous system, bones and joints, skin, eyes and endocrine glands [1]. Although dissemination is more common in immunocompromised individuals, disseminated coccidioidomycosis is also seen in immunocompetent individuals [1]. 

The main treatment for extra-thoracic, non-meningeal, *Coccidioides* infection relies on azole anti-fungal therapy [1]. The mechanism of action of anti-fungal azoles is a result of the inhibition of the enzyme 14-alpha-demethylase, which fungi need for the conversion of lanosterol to ergosterol, a key component of the fungal cell membrane [2]. The absence of ergosterol destabilizes the cell membrane, inhibiting cellular physiology and replication, and may result in lysis [2]. The most common agents used for the treatment of *Coccidioides* infection are fluconazole and itraconazole [1]. In refractory or quickly progressing non-meningeal disease, therapy may be switched to either posaconzaole, voriconazole or isavuconazole [3]. Clinicians must be aware of the potential side effects associated with prolonged anti-fungal azole therapy and their potential impact on the care of patients with disseminated coccidioidomycosis. In this report, we provide an example of an unusual presentation of iatrogenic glucocorticoid deficiency and pseudohyperaldosteronism secondary to posaconazole toxicity.

## 2. Case Summary

A 56-year-old man presented to our institution with a three-month history of nausea, vomiting, diffuse abdominal pain, and 30-pound unintentional weight loss. He was born in Mexico but had been living in the central valley of California for over 30 years working as a farmer harvesting broccoli. His past medical history included: stage 2 chronic kidney disease, hypertension, hyperlipidemia, gastroesophageal reflux disease and a two-year history of a left ankle destructive arthropathy secondary to chronic disseminated coccidioidomycosis. On admission, his medications included: daily amlodipine 5 mg, rosuvastatin 40 mg, furosemide 20 mg as needed, valsartan 80 mg, tamsulosin 0.4 mg and posaconazole 300 mg. For pain management, the patient was taking gabapentin 400 mg every 6 h. Five months prior to admission, his therapy was switched from fluconazole to posaconazole at the recommendation of his infectious disease physician owing to persistently elevated coccidioidal complement-fixation titer of 1:32 coupled with worsening pain in his left ankle. Two months prior to admission, the patient was evaluated by an orthopedic surgeon who ordered a 3-view left ankle radiograph. This imaging was consistent with end-stage destructive arthropathy, infectious in nature. At that time, he was deemed a non-surgical candidate by an orthopedic surgeon and posaconazole was continued.

One month prior to admission, the patient was admitted to another hospital on multiple occasions for abdominal pain, nausea, and vomiting, always associated with hypokalemia. He underwent an extensive workup, including multiple unremarkable CT abdomen and pelvis studies, a negative hepatobiliary scintigraphy scan, a negative cardiac stress test, and an esophagogastroduodenoscopy which was notable for mild gastritis with biopsy positive for *Helicobacter pylori*. It was thought at that time that his symptoms and hypokalemia were due to symptomatic *H. pylori* infection and vomiting, and he was discharged on triple therapy with clarithromycin, amoxicillin and pantoprazole. Unfortunately, his symptoms persisted, and he presented to our institution for further evaluation. 

On admission (day 0) to our hospital, his vital signs were: temperature 36.8 °C, pulse 73, blood pressure 129/92 mmHg, respiratory rate 18, oxygen saturation 100%. In addition to the weight loss and nausea, he reported muscle weakness, cramps, general malaise and fatigue. Physical examination was notable for a mildly ill-appearing man. He had dry mucous membranes, diffuse abdominal tenderness and left ankle tenderness to palpation without swelling or erythema. There was no evidence of temporal wasting, buccal or diffuse hyperpigmentation, or skin rash. His laboratory findings were significant for potassium 2.9 mmol/L, with no EKG changes, magnesium 1.5 mg/dL, and a serum creatinine of 1.14 mg/dL, which was his baseline. The remainder of his laboratory results are listed in Table 1. The patient’s urinalysis as well as both chest and abdominal radiographic studies were unremarkable. His transthoracic echocardiogram showed an ejection fraction of 60.3% without gross abnormalities. Review of his medical records from his hospitalization 2 weeks prior revealed a coccidioidal complement-fixation titer at 1:16, which was slightly decreased from 1:32 titer five months prior to admission. 

Given this patient’s constellation of symptoms, there was concern for glucocorticoid deficiency, though his hypokalemia was atypical for primary adrenal insufficiency associated with low mineralocorticoid activity. His initial random cortisol at 10:27 AM on day 0 was returned at 5.8 μg/dL (normal, >2.0). On day + 2, a morning Cosyntropin stimulation (one-time dose of 0.25 mg IV) test revealed an elevated ACTH level of 168 pg/mL (normal, 7.2–63.3), with a baseline serum cortisol of 4.6 μg/dL, which rose to only 7.2 μg/dL 60 min after administration (normal:cortisol ≥ 18 μg/dL 60 min after Cosyntropin administration), consistent with primary glucocorticoid deficiency. Aldosterone was not measured this admission, but his plasma renin activity level was suppressed to less than 0.6 ng/mL/h (normal, 0.6–4.3) with concurrent potassium of 3.4 mmoL/L, suggesting mineralocorticoid excess rather than deficiency. His serum posaconazole level was 5420 ng/mL (therapeutic range, 1000–3750 ng/mL), drawn 30 h after his last posaconazole dose, consistent with elevated posaconazole levels. 

Due to concern for the effects of posaconazole on his adrenal steroid hormone levels, the patient’s posaconazole therapy was stopped and was switched to fluconazole 800 mg PO daily on day + 3. For glucocorticoid deficiency, he was started on hydrocortisone 20 mg in the morning and 10 mg in the afternoon followed by taper to 10 mg in the morning and 5 mg in the afternoon as a maintenance dose. He had improvement of his gastrointestinal symptoms and electrolyte abnormalities on discharge. On follow-up 2 months after discharge, the patient’s fatigue and malaise had resolved, and he had regained 20 pounds. His potassium, renin, and aldosterone levels were measured and were all within normal limits. He had a repeat Cosyntropin stimulation test that showed ongoing primary glucocorticoid deficiency (Table 1), and hydrocortisone was continued at a maintenance dose. He is undergoing re-evaluation for surgical intervention for chronic coccidioidomycosis of his left ankle. 

Six months after his hospital admission, the patient had a follow-up visit with his primary endocrinologist at Stanford University Medical Center and had been receiving fluconazole, 600 mg daily for over 7 months. Repeat Cosytropin testing at this time revealed ongoing low cortisol levels despite normal ACTH (Table 1). The patient’s potassium, renin and aldosterone levels at this time were also normal (Table 1). Repeat CT scan of the abdomen showed normal appearance of both adrenal glands, and his serum 21-hydroxylase antibody testing was negative, suggesting no autoimmune involvement etiology.

## 3. Discussion

This case of iatrogenic glucocorticoid deficiency highlights key aspects in the management of patients with disseminated coccidioidomycosis on chronic anti-fungal azole therapy. Long-term use of anti-fungal azoles remains the therapy of choice for treating coccidioidomycosis [1]. In addition to medical therapy, source control remains critical. In cases where there is joint involvement, surgical debridement remains imperative [4]. It is critical to understand the potential side effects of anti-fungal azole therapy, as treatments can last a minimum of six months in patients with disseminated disease [4]. All anti-fungal azoles can cause hepatoxicity, ranging from mild transaminitis to fulminant hepatic failure [2]. Fluconazole is generally well-tolerated, but at high doses can lead to alopecia [5]. In the case of our patient, due to progression of his symptoms from his left ankle joint, his therapy was switched from fluconazole to posaconazole [4]. 

As with ketoconazole, posaconazole is known to interfere with steroid biosynthesis in human adrenal glands (Figure 1) [2]. Unique to posaconazole is the clinical presentation of hypokalemia and hypertension found in nearly a quarter of patients in a recent study [6]. This clinical picture of apparent mineralocorticoid excess secondary to posaconazole has been called posaconazole-induced pseudohyperaldosteronism (PIPH), and was clearly evident in our patient. Unlike the classical presentation of PIPH and chronic hypokalemia attributable to posaconazole, our patient also had profound glucocorticoid deficiency, which persisted despite switching to fluconazole. 

Ketoconazole is a potent inhibitor of several steps in the adrenal steroidogenesis pathway, including 20,22-desmolase, 17α-hydroxylase, 11β-hydroxylase, and 18-hydroxylase [7,8]. This side effect can be useful in some clinical settings of adrenal hormone excess, such as in paraneoplastic syndromes [9]. Recent clinical and molecular studies [10,11,12], have suggested that posaconazole inhibits 11β-hydroxylase (Figure 1) [10,11,13], thus decreasing production of aldosterone and cortisol from 11-deoxycorticosterone and 11-deoxycortisol, respectively. The accumulation of the aldosterone precursor 11-deoxycorticosterone has similar effects on the mineralocorticoid receptor and causes hypertension and promotes the excretion of potassium and absorption of sodium in the kidney distal tubules [10,11]. Levels of aldosterone or 11-deoxycorticosterone were not measured for this patient on admission, but we predict that he would have had low aldosterone and elevated 11-deoxycorticosterone, which is consistent with previous cases of pseudohyperaldosteronism. A second and likely concurrent mechanism proposed is that posaconazole inhibits the enzyme 11β-hydroxysteroid dehydrogenase type 2 (11βHSD2), which converts cortisol to cortisone in the distal renal tubule [11,12,13,14]. Cortisol also has potent mineralocorticoid activity and binds the receptor with high affinity. Inhibition of 11βHSD2 thus results in increased cortisol-mediated activation of mineralocorticoid receptors and contributes to pseudohyperaldosteronism [14]. In vitro data suggests that this effect may be more modest compared to the inhibition of 11β-hydroxylase by posaconazole [10]. 

To our knowledge, ours is the first case report to describe both glucocorticoid deficiency requiring hydrocortisone replacement together with PIPH. Most documented cases of PIPH have shown normal cortisol levels without signs of glucocorticoid deficiency [12,13,15]. One case report demonstrated mild adrenal insufficiency based on ACTH stimulation testing but was deemed not to require hydrocortisone replacement [11]. A second report described a patient who presented with nausea, malaise, hypokalemia, and a Cosyntropin test result similar to ours [16]. However, the authors attributed the hypokalemia to vomiting and primary mineralocorticoid deficiency rather than PIPH, and the patient was also prescribed both hydrocortisone and fludrocortisone; the latter would theoretically worsen hypokalemia and hypertension from PIPH. Repeat Cosyntropin testing in that particular case was normal one year after cessation of posaconazole, and hydrocortisone and fludrocortisone were discontinued at that time. Regarding our patient, although he presented with nausea and vomiting, the normalization of his renin levels and serum potassium levels once posaconazole was discontinued suggest that posaconazole was a major contributing factor to his hypokalemia.

Our patient continues to demonstrate poor glucocorticoid reserve seven months after cessation of posaconazole despite normal potassium, renin, and aldosterone levels, suggesting that the posaconazole-induced glucocorticoid deficiency is not as quickly reversible as it is for its effect on the mineralocorticoid pathway and receptor. It is also possible that his current use of fluconazole may also be causing adrenal insufficiency, which has been described in several case reports [17,18,19], though mostly in critically ill patients. Furthermore, in vitro studies on the impact of fluconazole on adrenal steroidogenesis inhibition have had variable results [20]. 

Nguyen et al. [6] demonstrated an association between posaconazole-induced pseudohyperaldosteronism and elevated serum posaconazole concentrations, which our patient had as well. It remains unclear as to why our patient developed such high serum levels of posaconazole. He had fluctuations in his renal function in the last few months preceding presentation, but several studies have demonstrated the safe use of posaconazole in patients with renal impairment, although the incidence of glucocorticoid deficiency was not evaluated [21,22]. Furthermore, the use of posaconazole is favored for the long-term treatment of disseminated coccidioidomycosis when fluconazole fails [23,24]. Careful review of the patient’s current medications did not reveal any drug-to-drug interactions that may have contributed to increased posaconazole levels. Older age and baseline hypertension have also been found to be associated with PIPH [7]. For our patient, his renal function remained normal months after cessation of posaconazole, as well as normal renin and aldosterone levels. Future studies advancing understanding of predisposing factors for posaconazole-induced adrenal insufficiency are warranted, especially as the use of this medication increases in the treatment of chronic fungal infections. Clinicians must also consider other azole alternatives in cases when posaconazole cannot be tolerated. For these patients, anti-fungal azoles including fluconazole, voriconazole or isavuconazole have been suggested to have less affinity to 11β-hydroxylase or 11βHSD2, making them better candidates for chronic treatment of coccidioidomycosis [10].

## 4. Conclusions

Glucocorticoid deficiency may present concurrently with apparent mineralocorticoid excess with posaconazole therapy. It is imperative for clinicians to be aware of the effects of posaconazole on adrenal steroid hormone pathways. The treatment of disseminated coccidioidomycosis can be challenging, and posaconazole is still appropriate in cases where there is progression or poor response to fluconazole. Therapeutic drug monitoring of posaconazole levels may be considered in patients requiring long-term therapy and would alert clinicians to undesirable side effects or blood concentrations insufficient for adequate response. ACTH stimulation testing may be warranted in all patients with high serum posaconazole levels, signs of pseudohyperaldosteronism (i.e. hypertension and/or hypokalemia), or symptoms of glucocorticoid deficiency, with long-term treatment with glucocorticoids as necessary.

## Figures and Tables

**Figure 1 jof-07-00620-f001:**
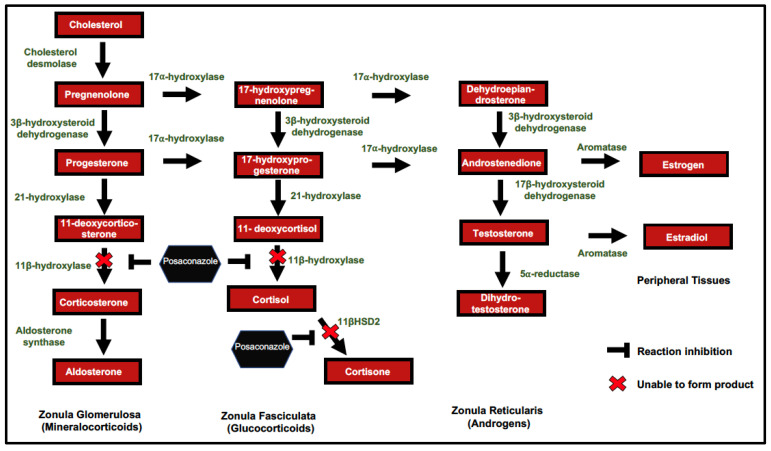
Potential impact of posaconazole in human steroid synthesis. 11β-hydroxysteroid dehydrogenase type 2 (11βHSD2).

**Table 1 jof-07-00620-t001:** Summary of laboratory results.

Lab Name (Units)	Initial Presentation	3 Months after Posaconazole Cessation	7 Months after Posaconazole Cessation	Reference Range
Potassium (mmol/L)	2.9	4.1	4.7	3.5–5.5
ACTH (pg/mL)	168 pg/mL	123	34	7.2–63.3
Cortisol, Baseline (μg/dL)	5.8	3.1	0.2	≥2.0
Cortisol,1-h post-Cosyntropin (μg/dL)	7.2	4.1	0.8	≥18
Renin Activity (ng/mL/h)	<0.6	1.2	2.6	0.6–4.3
Aldosterone (ng/dL)	-	8.6	<4.0	<21
Posaconazole (ng/mL)	5420	-	-	1000–3750

Serum cortisol levels were determined via antibody testing. Serum aldosterone was measured via liquid chromatography and mass spectrometry.

## Data Availability

No new data were created or analyzed in this study. Data sharing is not applicable to this article.

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
