# Peer review of "Severe Posaconazole-Induced Glucocorticoid Deficiency with Concurrent Pseudohyperaldosteronism: An Unfortunate Two-for-One Special"

_jof, 2021, doi:10.3390/jof7080620_

Round 1

Reviewer 1 Report

The work presented to me for review concerns a very serious problem of adrenal insufficiency during the use of antifungal drugs. When prescribing this type of medication, most doctors do not remember about such severe side effects. Adrenal insufficiency is one of the most serious life-threatening conditions. Failure to activate hydrocortison at the right time can result in loss of life. The described case shows a 56-year-old man treated with posaconazole for coccidiomycosis. During treatment, the patient presented typical symptoms of Addison's disease. The presented case presents insufficiency of the adrenal cortex and pseudohyperaldosteronism, which appear during treatment with antifungal preparations.

The work has a typical layout for a case presentation. I have no major substantive comments. In the summary chapter, it is worth mentioning that steroidogenesis inhibitors are also used in the treatment of hypercortisolemia caused by the secretion of paraneoplastic ACTH.

Author Response

Dear Reviewer 1

We greatly appreciate your comments and suggestions. Regarding your comments, please see our response below:

“The work presented to me for review concerns a very serious problem of adrenal insufficiency during the use of antifungal drugs. When prescribing this type of medication, most doctors do not remember about such severe side effects. Adrenal insufficiency is one of the most serious life-threatening conditions. Failure to activate hydrocortison at the right time can result in loss of life. The described case shows a 56-year-old man treated with posaconazole for coccidiomycosis. During treatment, the patient presented typical symptoms of Addison's disease. The presented case presents insufficiency of the adrenal cortex and pseudohyperaldosteronism, which appear during treatment with antifungal preparations.

The work has a typical layout for a case presentation. I have no major substantive comments. In the summary chapter, it is worth mentioning that steroidogenesis inhibitors are also used in the treatment of hypercortisolemia caused by the secretion of paraneoplastic ACTH.”

Response:

We have incorporated your suggestion in our discussion in our revised manuscript in how the use of agents such as ketoconazole can be used as a steroid hormone blocker.

Reviewer 2 Report

This paper is a case report confirming prior reports that Posaconazole can cause a  pseudohyperaldosteronism syndrome, characterized by hypokalemia, hypertension, depressed renin, very low plasma aldosterone and elevated plasma levels of 11-deoxycortisol and or 11-deoxycorticosterone.  This case report had nausea and vomiting as another cause of hypokalemia, no hypertension, no quantitation of how low renin activity was, and no measurement of 11-deoxycortisol  or 11-deoxycorticosterone. The paper also reports this patient had decreased adrenal function, manifest as elevated plasma ACTH and low response to Cosyntropin. Plasma cortisol was normal. Dose of Cosyntropin was not given.  Although the authors concluded that the adrenal insufficiency continued, the patient remained on adrenal replacement with hydrocortisone. Replacment can suppress adrenal response to Cosyntropin, depending on dose and duration. Hydrocortisone also appears in the plasma measurement of cortisol. If one wishes to measure cortisol during replacement therapy, dexamethasone is preferred as replacement.

 The conclusion of long term adrenal suppression needs better documentation, as does the diagnosis of pseudohyperaldosteronism.

 Table 1: Would be better to categorize column headings as 3 and 7 months after Posaconazole “cessation” for clarity.

Line 39: please remove “the” before evident

Line 142: please change “persistent” to persisted

Author Response

Dear Reviewer 2

We greatly appreciate your comments and suggestions. Regarding your comments, please see our response below:

“This paper is a case report confirming prior reports that Posaconazole can cause a pseudohyperaldosteronism syndrome, characterized by hypokalemia, hypertension, depressed renin, very low plasma aldosterone and elevated plasma levels of 11-deoxycortisol and or 11-deoxycorticosterone. This case report had nausea and vomiting as another cause of hypokalemia, no hypertension, no quantitation of how low renin activity was, and no measurement of 11-deoxycortisol or 11-deoxycorticosterone. The paper also reports this patient had decreased adrenal function, manifest as elevated plasma ACTH and low response to Cosyntropin. Plasma cortisol was normal. Dose of Cosyntropin was not given. Although the authors concluded that the adrenal insufficiency continued, the patient remained on adrenal replacement with hydrocortisone. Replacment can suppress adrenal response to Cosyntropin, depending on dose and duration. Hydrocortisone also appears in the plasma measurement of cortisol. If one wishes to measure cortisol during replacement therapy, dexamethasone is preferred as replacement.

The conclusion of long term adrenal suppression needs better documentation, as does the diagnosis of pseudohyperaldosteronism.

Table 1: Would be better to categorize column headings as 3 and 7 months after Posaconazole “cessation” for clarity.

Line 39: please remove “the” before evident

Line 142: please change “persistent” to persisted”

Response:

Thank so much for your thoughtful comments and suggestions. In the regard to the patient’s blood pressure upon initial presentation, he was taking multiple blood pressure medications upon admission including amlodipine 5 mg and valsartan 80 mg. In addition, his poor intake upon admission may have also contributed to his normotension. We unfortunately do not know how long he has been on these medications or if his doses were increased after starting his posaconazole treatment. Upon review of other case reports, including the one by Miller et al. (reference 16 in the revised manuscript), the presented patient was also normotensive with a BP 125/60 mmHg. This suggests that hypertension may not always be present on presentation for patients in which we suspect pseudohyperaldosteronism syndrome.

We did obtain the initial renin activity level outlined in Table 1, which was less than 0.6 ng/mL/hr (reference range 2.9 -10.8), indicating that it was suppressed. As pointed out, we sadly do not have the patient’s aldosterone levels nor 11-deoxycortisol or 11-deoxycorticosterone (DOC) levels. Having these values would’ve been useful in supporting the diagnosis of pseudohyperaldosteronism. Based on the prior literature of PIPH and known mechanisms of action of posaconazole on the 11-beta hydroxylase and 11-beta HSD2, we presume here that aldosterone would have been low (and 11-deoxycortisol and DOC would have been high) but having these levels at the time would have been helpful in supporting the diagnosis. We have included this statement as well in the discussion portion of the revised manuscript.

In respect to the patient’s plasma cortisol levels on presentation, we agree that 5.8 ug/dL still falls within the normal range. In accordance to the American College of Physicians guidelines, if AM cortisol levels range from 3-15 ug/dL, it is reasonable to proceed with a ACTH stimulation test, which is what we did given our high clinical suspicion of adrenal insufficiency. The dose of Cosyntropin given for his initial test was 0.25 mg IV. We have added this to the revised manuscript. The fact that the patient did not respond appropriately to Cosyntropin stimulation confirms adrenal insufficiency.

It is possible that the patient’s nausea and vomiting may had a been a contributing factor to his hypokalemia, as outlined in the case presentation. These signs and symptoms were present months prior to admission to Stanford University Hospital. However, subsequent potassium testing while patient was on hydrocortisone therapy demonstrated normal levels, suggesting that decreased cortisol levels were a main contributor to his particular case. We also addressed this on lines 186 to 195 of the original and revised manuscript.

Steroid therapy can blunt the endogenous ACTH elevation over the time, and this is what we saw by 3 months and 7 months after posaconazole cessation. However, if steroids are given at physiologic doses (e,g. hydrocortisone 15-20 mg/day), the hypothalamus-pituitary-adrenal (HPA) axis may be adequately stimulated to eventually recover and allow sufficient endogenous cortisol production from the adrenal glands. We prefer to use hydrocortisone for steroid tapering due its short half-life, allowing stimulation of the HPA axis between doses. The patient was on a physiologic maintenance hydrocortisone dosage (10 mg in AM and 5 mg in PM).

Prior to outpatient Cosyntropin stimulation testing, the patient was told to hold the morning hydrocortisone prior to the test as well as the afternoon hydrocortisone the day before the test to ensure that no hydrocortisone is circulating to affect the cortisol level. The patient confirmed that he did so.

We prefer not to use dexamethasone as the preferred chronic replacement strategy due to its long half-life and strong potency. Dexamethasone has a higher risk of the development of Cushingoid features over time, and it would need to be held for for >24-48 hours prior to Cosyntropin stimulation testing. Dexamethasone may be used in acute adrenal crisis when a glucocorticoid should be given emergently while also performing a Cosyntropin stimulation test to confirm a diagnosis of adrenal insufficiency.

We have made the edits recommended for the Table 1 header and have edited the lines 139 (which corresponds to your suggestion, and not line 39) as well as in line 142 of the revised manuscript.

Reviewer 3 Report

Authors describe a very interesting case of posaconazole-induced pseudohyperaldosteronism along with a severe glucocorticoid deficiency. It is an important case raising awareness of endocrinologists and infectiologists for adverse effects by posaconazole.

I only have a few minor comments:

  • Line 37: change “azole” to “azole anti-fungal” and “of azoles” to “of anti-fungal azoles”. It is a detail but there are many azoles not used for anti-fungal therapy. (also line 44 and in the discussion)
  • How has cortisol and aldosterone been quantified, using antibody-based method or using LC-MS? Include a footnote in Table1 mentioning the method.
  • The low cortisol after discontinuation of posaconazole is highly atypical. Verify that indeed hydrocortisone (cortisol) been used and not prednisone or prednisolone for treating glucocorticoid deficiency. If prednisone/prednisolone were used, this would explain low cortisol after 7 months. With that low cortisol concentration of 0.2 it is surprising the patient’s fatigue and metabolic problems resolved after discontinuation and administration of hydrocortisone (which seems not be detectable?). It is also surprising if the patient is given cortisol that cortisol is not detected, which may indicate a rapid hepatic glucocorticoid metabolism, i.e. elevated 5alpha-reductase and 5beta-reductase along with elevated cytochrome P450 dependent glucocorticoid metabolism. But then again one would expect insufficient glucocorticoid receptor activity and ACTH should be elevated (after the 7 months assessment). HPA axis might be impaired. However, ACTH was high when the patient first was studied.
  • Line 159: change “strong affinity” to “high affinity”
  • Consider in conclusions to recommend therapeutic drug monitoring. If serum posaconazole levels would be kept below 3 mg/L risk for adverse effects can be reduced, although likely not in all patients.

Author Response

Dear Reviewer 3

We greatly appreciate your comments and suggestions. Regarding your comments, please see our response below:

“I only have a few minor comments:

  • Line 37: change “azole” to “azole anti-fungal” and “of azoles” to “of anti-fungal azoles”. It is a detail but there are many azoles not used for anti-fungal therapy. (also line 44 and in the discussion)
  • How has cortisol and aldosterone been quantified, using antibody-based method or using LC-MS? Include a footnote in Table1 mentioning the method.
  • The low cortisol after discontinuation of posaconazole is highly atypical. Verify that indeed hydrocortisone (cortisol) been used and not prednisone or prednisolone for treating glucocorticoid deficiency. If prednisone/prednisolone were used, this would explain low cortisol after 7 months. With that low cortisol concentration of 0.2 it is surprising the patient’s fatigue and metabolic problems resolved after discontinuation and administration of hydrocortisone (which seems not be detectable?). It is also surprising if the patient is given cortisol that cortisol is not detected, which may indicate a rapid hepatic glucocorticoid metabolism, i.e. elevated 5alpha-reductase and 5beta-reductase along with elevated cytochrome P450 dependent glucocorticoid metabolism. But then again one would expect insufficient glucocorticoid receptor activity and ACTH should be elevated (after the 7 months assessment). HPA axis might be impaired. However, ACTH was high when the patient first was studied.
  • Line 159: change “strong affinity” to “high affinity”
  • Consider in conclusions to recommend therapeutic drug monitoring. If serum posaconazole levels would be kept below 3 mg/L risk for adverse effects can be reduced, although likely not in all patients.”

Response:

We would like to thank you for taking the time to review our manuscript. In response to your comments:

We have corrected azole to azole antifungal in line 37 as recommended. We have also changed this in the discussion of the revised manuscript as recommended.

At Stanford University Hospital, serum cortisol is measured via antibody testing. For serum aldosterone, this test is sent to the Mayo Clinic, for which they perform LC-MS. This has also been added to the footnote of Table 1.

We can verify that he was indeed taking hydrocortisone and not prednisone or prednisolone. The baseline cortisol and 1-hour post-Cosyntropin cortisol measurements at 3 months and 7 months after posaconazole cessation are in the context of holding hydrocortisone that morning and the day before (see above) to assess his HPA axis status. We agree that it is surprising how low his cortisol levels are. We would have expected higher levels after not receiving posaconazole for this duration. It is possible that fluconazole may also be causing some adrenal insufficiency as we speculated in the paper. If this were the case, his ACTH should be more elevated but perhaps his chronic hydrocortisone replacement has lowered the ACTH response over time (also as described above).

We have changed “strong affinity” to “high affinity” as recommended. This can be noted on the revised manuscript and added to our conclusion paragraph the recommendation for drug monitoring in our revised manuscript.

Reviewer 4 Report

A well presented and well written case report, educative for those to read.

A question only: Was during patient's last hosptalization the first time after six months of therapy that levels of posaconazole were measured? Should we have TDM in case of chronic treatment  with posaconazole (at least during the first month of treatment?). If yes. this should be mentioned and advised

Author Response

Dear Reviewer 4

We greatly appreciate your comments and suggestions. Regarding your comments, please see our response below:

“A well presented and well written case report, educative for those to read.

A question only: Was during patient's last hospitalization the first time after six months of therapy that levels of posaconazole were measured? Should we have TDM in case of chronic treatment with posaconazole (at least during the first month of treatment?). If yes. this should be mentioned and advised”

Response:

We would like to thank you for the kind remarks. Unfortunately, we do not have any records of whether posaconazole levels were tested during the patient’s previous hospitalization. As described in the manuscript, it seems that the physicians were favoring H. pylori gastritis as the culprit. We do agree that drug monitoring of posaconazole should be considered, especially in patients who will require long term treatment. We have included this in the conclusion section of our revised manuscript.

Round 2

Reviewer 2 Report

The fundamental flaw is the data do not support the authors' conclusions. No editing can correct this problem.